# Detection of β-Lactamase-Producing *Enterococcus faecalis* and Vancomycin-Resistant *Enterococcus faecium* Isolates in Human Invasive Infections in the Public Hospital of Tandil, Argentina

**DOI:** 10.3390/pathogens9020142

**Published:** 2020-02-20

**Authors:** Celia M. Schell, Ana P. Tedim, Mercedes Rodríguez-Baños, Mónica D. Sparo, Sabina Lissarrague, Juan A. Basualdo, Teresa M. Coque

**Affiliations:** 1Centro Universitario de Estudios Microbiológicos y Parasitológicos (CUDEMyP), Centro Universidad Nacional de La Plata asociado a Comisión de Investigaciones Científicas (CIC), Facultad de Ciencias Médicas, Universidad Nacional de La Plata. Av. 60 y 120 s/n, 3er piso, CP 1900 La Plata, Buenos Aires, Argentina or monicasparo@gmail.com (M.D.S.); jabasua@med.unlp.edu.ar (J.A.B.); 2Servicio de Microbiología, Hospital Universitario Ramón y Cajal, Instituto Ramón y Cajal de Investigación Sanitaria (IRYCIS), Carretera de Colmenar, km. 9.1, Planta -1IZQ, 28034 Madrid, Spain or assantos.iecscyl@saludcastillayleon.es (A.P.T.); merche1976es@yahoo.es (M.R.-B.); 3Centros de Investigación Biomédica en Red de Epidemiología y Salud Pública (CIBER-ESP), Av. Monforte de Lemos, 3-5. Pabellón 11. Planta 0, 28029 Madrid, Spain; 4Laboratorio de Microbiología Clínica, Hospital Municipal Ramón Santamarina, Gral. Paz 1406, B7000 Tandil, Buenos Aires, Argentina; 5Unidad de Resistencia a Antibióticos y Virulencia Bacteriana asociada al Consejo Superior de Investigaciones Científicas (CSIC), 28006 Madrid, Spain

**Keywords:** *Enterococcus faecalis*, *Enterococcus faecium*, invasive infections, antibiotic resistance, VRE, *bla*^+^

## Abstract

The study’s aim was to analyze the population structure of enterococci causing human invasive infections in a medium-sized Argentinian Hospital coincidental with a 5 year-period of increased recovery of antibiotic resistant enterococci (2010–2014). Species identification (biochemical testing/MALDI-TOF-MS), antimicrobial susceptibility (disk-diffusion) and clonal relatedness (PFGE/MLST/BAPS) were determined according to standard guidelines. β-lactamase production was determined by a nitrocefin test and confirmed by PCR/sequencing. The isolates were identified as *Enterococcus faecalis* and *Enterococcus faecium* at a 2:1 ratio. Most of the *E. faecalis* isolates, grouped in 25 PFGE-types (ST9/ST179/ST236/ST281/ST388/ST604/ST720), were resistant to high-levels (HLR) of gentamicin/streptomycin. A ST9 clone (*bla*^+^/HLR-gentamicin) was detected in patients of different wards during 2014. *E. faecium* isolates were grouped in 10 PFGE-types (ST25/ST18/ST19/ST52/ST792), with a low rate of ampicillin resistance. Five vancomycin-resistant *E. faecium,* three *vanA* (ST792/ST25) and two *vanB* (ST25) were detected. The ST25 clone carried either *vanA* or *vanB*. The recovery of a *bla*^+^-ST9-*E. faecalis* clone similar to that described in the late 1980s in Argentina suggests the possibility of a local hidden reservoir. These results reflect the relevance of local epidemiology in understanding the population structure of enterococci as well as the emergence and spread of antimicrobial resistance in predominant enterococcal clonal lineages.

## 1. Introduction

*Enterococcus faecalis* and *Enterococcus faecium* became two of the most important nosocomial pathogens in recent decades [1]. The treatment of severe enterococcal infections is frequently impaired by the intrinsic and/or acquired resistance to first-line antibiotics, namely, those active against the cell wall (β-lactam or glycopeptides) and aminoglycosides, which combine to achieve a bactericidal effect [1]. Resistance to these therapeutic choices has been extensively reported in Western countries [2,3] but information from other locations including Argentina is still scarce and comes from studies focused on glycopeptide resistance [4,5]; early descriptions of emerging mechanisms of resistance (β-lactamase production) [6]; or cross-sectional surveillance studies which only include a few isolates from different geographical locations (https://resistancemap.cddep.org/, [7]).

Enterococci are intrinsically resistant to some β-lactam antibiotics such as cephalosporins and carbapenems but resistance to penicillin is acquired either by mutations in penicillin binding proteins (PBPs) or, less frequently, by the production of a β-lactamase [8]. Resistance to aminopenicillins is very common in *E. faecium* and is mostly due to mutations in the PBP5 [9], and sporadically, to the production of β-lactamase [10]. Although most *E. faecalis* isolates are susceptible to penicillin, penicillin-resistant and ampicillin-susceptible *E. faecalis* (PRASEF) have been reported since the late 1980s in different countries including Argentina [11,12]. To date, PRASEF can result from the production of β-lactamase or mutations in the PBP4 [13]. Resistance to glycopeptides is mediated by a plethora of genetic determinants, with genotypes *vanA* (Tn*1546*) and *vanB* (Tn*5382*/Tn*1547*) being the most predominant [3]. The first vancomycin-resistant enterococci (VRE) reported in Latin-America was isolated in Mendoza, Argentina, in 1996 from a 7-year old male patient treated with different antibiotics and was identified as *E. faecium* (*vanA*) [14]. After this sporadic case, *E. faecium* harbouring *vanA* or *vanB* in colonized or infected patients were detected in several Argentinean hospitals [4,5,15]. Most of these VRE were *E. faecium* (*vanA*) [3,16] and, sporadically, *E. faecalis* and *Enterococcus gallinarum* [3,17].

High-level resistance (HLR) to gentamicin in enterococci was first described in 1979 in France and by the late 1980s, in different countries including Argentina which is often associated with *E. faecalis* β-lactamase producers [11,12]. Recent data at the CDDEP site (https://resistancemap.cddep.org/) revealed that Argentinian *E. faecium* invasive isolates are commonly resistant to ampicillin (>75%–80%) and vancomycin (60%–75%) while *E. faecalis* invasive isolates are rarely resistant to cell-wall active antibiotics. HLR to gentamicin is often observed in both species (60% and 30%, respectively).

Most *E. faecium* and *E. faecalis* clinical isolates belong to a few sequence types (STs), namely ST6, ST9, ST16 and ST87 for *E. faecalis*, and ST17, ST18 and ST78 for *E. faecium*, some of them overrepresented in different geographical areas [2,18]. For *E. faecium*, population structure often combines MLST and Bayesian analysis of the population structure (BAPS). Hospital isolates often cluster in BAPS subgroups 2.1a (ST117, ST203 and ST80) and 3.3a (ST18 and ST17) while community-based isolates belong to BAPS subgroups 1.2 and 3.3b [18,19]. These predominant STs are also called “high-risk clonal complexes” [20].

The aim of this study was to characterize the *E. faecalis* and *E. faecium* from human invasive infections in the Public Hospital of Tandil, Argentina, a medium-sized hospital covering urban and rural human populations. The interest of the study lies in its value for describing the population structure of enterococci during a period of increasing recovery of multidrug-resistant (MDR) isolates in a geographical area with low prevalence of enterococci resistance to first-line antibiotics but where emblematic mechanisms of resistance were detected early.

## 2. Results

### 2.1. Epidemiological Background of the Strains Isolated from Human Invasive Infections in Hospital Ramón Santamarina (HRS).

Epidemiological data of the 63 *Enterococcus* spp. strains (44 *E. faecalis* and 19 *E. faecium*) analysed in this study are shown in Table 1 and Table 2. Both *E. faecalis* and *E. faecium* were isolated from seven samples (three peritoneal fluids, two liver abscesses, one abdominal fluid, and one synovial fluid). The age of the patients ranged from 16 to 92 years (59 ± 18.8-y, 70% > 50 years old), most of them with an underlying disease (49%) and a history of antibiotic exposure (80%), mainly to ciprofloxacin (23.6%), cephalexin (18.2%) and ceftriaxone (14.5%). The mortality rate in this series was 27.3% (Appendix A).

### 2.2. E. faecalis

Approximately half of the *E. faecalis* isolates (47.7%) were susceptible to all antibiotics tested (Figure 1). HLR to gentamicin (43.2%), streptomycin (22.7%) or both (13.6%) and resistance to fluoroquinolones (20.4%, ciprofloxacin and levofloxacin), penicillin (11.4%) and chloramphenicol (2.3%) were detected. The production of β-lactamase was inferred for the five PRASEF isolates based on the 5 mm increase in the inhibition diameter to ampicillin-sulbactam compared to ampicillin [21], a positive nitrocefin test and the identification of a class A β-lactamase gene conferring resistance to aminopenicillins (GenBank accession number U43087.1). None of these strains showed mutations in the PBP4 previously associated with possible penicillin resistance (data not shown). The PRASEF isolates exhibited the same PFGE-type, EFC-2, and were classified as ST9. The rest of the *E. faecalis* strains were grouped in 24 different PFGE-types. Besides EFC-2, the most common PFGE-types were EFC-7, EFC-16 and EFC-3 which correspond to ST179, ST281, and ST720, respectively. ST720 is a novel *E. faecalis* ST described here for the first time (Table 2).

### 2.3. E. faecium

*E. faecium* isolates were resistant to penicillin (47.4%), ampicillin and ampicillin/sulbactam, vancomycin (26.3% each), teicoplanin, levofloxacin and quinupristin-dalfopristin (15.8% each), ciprofloxacin and high levels of gentamicin (10.5%; Figure 1). Only one isolate showed HLR to streptomycin. All *E. faecium* strains were susceptible to linezolid, tigecycline and chloramphenicol. Three *E. faecium* strains were MDR according to Magiorakos et al. (phenotypic resistance to three or more antibiotic families) [22]. *E. faecium* strains were grouped in 10 different PFGE-types (Figure 2), the predominant ones being: EFM-1, EFM-2 and EFM-4. EFM-1 and EFM-4 belonged to ST25-BAPS 2.3 and ST52-BAPS 7, respectively.

Five vancomycin-resistant *E. faecium* strains (three *vanA* and two *vanB*) were detected in this study. Two *vanA*
*E. faecium* isolates had different PFGE-types (EFM-7 and EFM-9) but both belonged to BAPS 3.1-ST792. The other *vanA* strain and the two *vanB* strains showed the same PFGE-type, EFM-1, and were identified as BAPS 2.3-ST25. The two *vanB* strains were isolated from blood and abdominal fluid samples of patients with documented bloodstream and intra-abdominal infections at the surgery and ICU wards in 2013 and 2014. 

## 3. Discussion

This report documents the presence of relevant high-risk clonal complexes of *E. faecalis* and *E. faecium* [2], underrepresented in most of the studies in Western countries but able to acquire and disseminate resistant genes to first-line antibiotics.

Among *E. faecalis*, the ST9-PRASEF clone (*bla*^+^/HLR-gentamicin) identified in this study represents one of the few *bla*^+^-*E. faecalis* strains described to date, most of them documented in the late 1980s in the USA, Lebanon, Canada and Argentina [23]. The apparent relationship between the ST9-*bla*^+^ isolates described here and those reported in another hospital of Buenos Aires in 1989, both showing HLR to gentamicin, suggest that this clone could have been circulating in our area since the late 1980s. Geographical endemicity of *E. faecalis* with infrequent mechanisms of resistance, such as the production of β-lactamase or resistance to vancomycin, have previously been described in specific regions of the USA, either due to an epidemic clone (ST6-*bla*^+^ ) [11,23] or an epidemic plasmid (Inc18-vanA) [24]. To date, it is not well understood why these antibiotic resistant strains remain apparently confined to specific regions. The presence of other *E. faecalis* such as ST179, ST388 and ST720 (HLR-gentamicin) in more than one patient in different wards reflects the transmissibility of several clones in our hospital.

Similarly, the *E. faecium* strains did not belong to clonal groups predominant in most hospitals as BAPS subgroups 3.3a (ST18 and ST17) or 2.1a (ST117, ST203 and ST80) [18,19,25,26]. Instead, it is of note that the detection of clones of other phylogenomic groups, often associated with animals and able to acquire different resistance traits such as BAPS 3.1-ST792 (2 *vanA*) or BAPS 2.3-ST25 (2 *vanB* and 1 *vanA*) [18,19,27]. This clonal diversity explains the low occurrence of ampicillin resistant *E. faecium* found in our study in comparison with that reported in other series (26.3% vs. >85%) (https://resistancemap.cddep.org/CountryPage.php?countryId=65&country=Argentina, [16,17,28]).

The diversity of *E. faecalis* and *E. faecium* able to acquire genes encoding HLR to gentamicin and streptomycin, some clones with zoonotic potential, might facilitate the spread of these genes between different hosts, as recently reported in our area [19,25,28,29].

Despite the limited sample analysed, epidemiological data of this series, the *E. faecalis*:*E. faecium* prevalence ratio, the diversity of clinical presentations [2,18,25], the age/sex of the patients and the risk factors for the acquisition of enterococcal infections [18,30] were in agreement with other studies.

In summary, the epidemiology of enterococci in a medium-sized hospital in South America during a non-outbreak situation revealed interesting information for public health. The persistence of emblematic and unusual resistant clones such as *E. faecalis* ST9 (*bla*^+^, HLR-gentamicin) suggests the presence of hidden reservoirs for MDR *E. faecalis* in different geographical areas. Moreover, it highlights the importance of defining the population structure of enterococci in different locations in order to understand the influence of sociodemographic factors in the clonal diversity of enterococci and thus in the emergence and transmission of antimicrobial resistance.

## 4. Materials and Methods

### 4.1. Epidemiological Data and Sampling of Enterococcus spp.

We retrospectively analysed all *Enterococcus* spp. strains consecutively isolated from patients with clinically documented invasive infections who were hospitalized at the HRS between 2010 and 2014. The HRS is a medium-sized hospital of 120 beds that provides specialized attention to a population size of ~130,000 habitants in the area of Buenos Aires (Argentina). More precisely, 41,000 individuals were attended to at the HRS during the period 2013–2014. After the study, only a few isolates resistant to first-line antibiotics were recorded (data not shown).

The samples analysed included blood (*n* = 22), abscess (*n* = 12), synovial fluid (*n* = 7), abdominal fluid (*n* = 6), peritoneal fluid (*n* = 6), intravesicular fluid (*n* = 1) and pericardial fluid (*n* = 1). One colony per morphology per patient was selected for further studies. All strains were identified with biochemical conventional tests [31] and confirmed by MALDI-TOF-MS (Bruker Daltonics, Bremen, Germany).

### 4.2. Ethical Approval

Patient records (underlying diseases, previous antimicrobial therapy, mortality, age and gender) were obtained in compliance with National Law No. 25.326 art. 11 of “Personal Data Protection” and National Law No. 26529/10 “Patient Rights, Clinical History and Informed Consent” of Argentina, in line with the Helsinki statement. A computerized data system was implemented at the HRS to optimize the management of information through the Integrated System of Argentinian Sanitary Information (SISA) in 2011. Due to the lack of this computerized data system, it was not possible to obtain all the data from the clinical history of some patients, especially those deceased.

### 4.3. Antimicrobial Susceptibility

Susceptibility to 13 antibiotics was determined by the disc diffusion method and using the ADAGIO™ Automated System (Bio-Rad, Hercules, CA, USA) as described. The antibiotics tested included ampicillin (10 µg), penicillin (6 µg), ampicillin/sulbactam (20 µg), chloramphenicol (30 µg), vancomycin (5 µg), teicoplanin (30 µg), streptomycin (300 µg), gentamicin (120 µg), ciprofloxacin (5 µg), levofloxacin (5 µg), quinupristin-dalfopristin (15 µg), linezolid (30 µg) and tigecycline (15 µg) (Bio-Rad, Hercules, CA, USA)]. Susceptibility to aminoglycosides, glycopeptides, quinolones and β-lactam antibiotics was also determined by an E-test (M.I.C. Evaluator™, OXOID, Basingstoke, UK). The methods and the interpretation of the results followed the CLSI guidelines [32]. *Enterococcus faecalis* ATCC 29212 and *Staphylococcus aureus* ATCC 25923 were used as control strains.

### 4.4. β-lactamase Production

β-lactamase production was preliminary tested by the nitrocefin test (BD BBL, Franklin Lakes, NJ, USA), according to the manufacturer’s instructions and further confirmed by PCR, and sequencing [33].

### 4.5. Detection of van Genes

*van* genes were detected by a multiplex PCR assay as previously described [34,35].

### 4.6. Clonal Relatedness

Clonal relatedness was preliminarily established by Pulsed Field Gel Electrophoresis (PFGE) as previously described [36,37]. A representative isolate per PFGE-type was further characterized by multi-locus sequence typing (MLST) according to PubMLST guidelines (http://pubmlst.org/) [38,39]. *E. faecium* MLST data were further characterized using by BAPS [18,19].

### 4.7. Statistical Analysis

Differences in the prevalence of tested features in *E. faecalis* and *E. faecium* strains were assessed by Chi-square and Fisher tests. A significance was established at *p* < 0.05.

## Figures and Tables

**Figure 1 pathogens-09-00142-f001:**
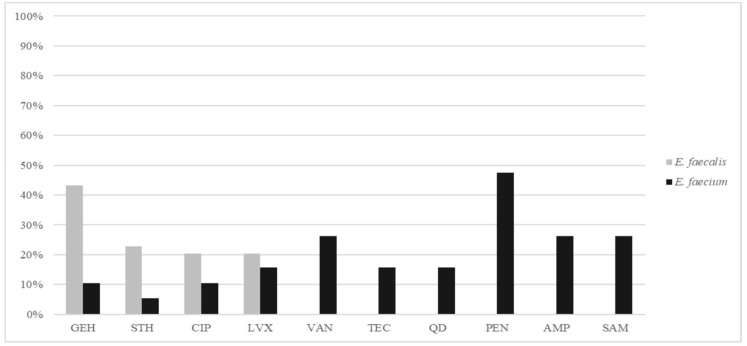
Percentage of antimicrobial resistance in *E. faecalis* and *E. faecium* strains isolated from invasive infections.

**Figure 2 pathogens-09-00142-f002:**
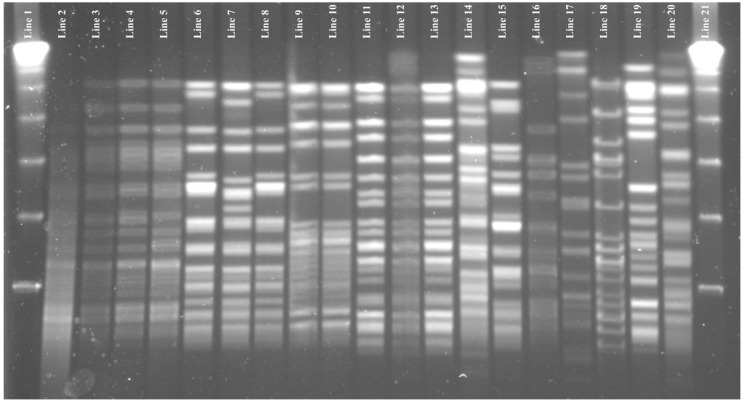
PFGE patterns of *Sma*I-digested genomic DNA of *E. faecium* strains analysed in this study. Line 1 to 21: λ, lambda ladder Marker. Line 2: C30-IRE1 *vanB*^+^; line 3: C38-IRE1; line 4: C47-IRE2 *vanB*^+^; line 5: C48-IRE1 *vanA*^+^; line 6: C31-IR; line 7: C32-IR; line 8: C34-IR; line 9: C45-IR; line 10: C52-IRE1; line 11: C16-IRE1; line 12: C20-IRE2; line 13: C35-IR; line 14: C7-IRE2; line 15: C14-IRE2; line 16: C40-IRE1; line 17: C44-IRE2 *vanA*^+^; line 18: C51-IRE2; line 19: C53-IR *vanA*^+^; line 20: C3-IR.

**Table 1 pathogens-09-00142-t001:** Relevant characteristics of the 44 *E. faecalis* strains isolated from human invasive infections at the Hospital Ramón Santamarina (HRS), Buenos Aires (Argentina).

Strain	Clinical Sample	Year	Antibiotic Susceptibility	PFGE-Type/ β-lactamase^+^	ST	HRS Ward
C43-IR	Renal abscess	2014	GEH, PEN	EFC-2/*bla*^+^	9	Surgery
C46-IR	Abdominal fluid	2014	GEH, PEN	EFC-2/*bla*^+^	9	Surgery
C47-IRE1	Abdominal fluid	2014	GEH, PEN	EFC-2/*bla*^+^	9	Surgery
C50-IR	Blood	2014	GEH, PEN	EFC-2/*bla*^+^	9	IM
C51-IRE1	Peritoneal fluid	2014	GEH, PEN	EFC-2/*bla*^+^	9	Surgery
C11-IR	Abdominal fluid	2010	GEH, STH, CIP, LVX	EFC-3	720	Unknown
C13-IR	Blood	2013	STH, CIP, LVX	EFC-3	720	Unknown
C22-IR	Blood	2013	GEN, STH, CIP, LVX	EFC-3	720	Unknown
C33-IR	Abscess	2013	GEH, STH, CIP, LVX	EFC-3	720	ER
C54-IR	Blood	2014	GEH	EFC-4	388	Traumatology
C55-IR	Blood	2014	GEH	EFC-4	388	ICU
C12-IRE1	Blood	2013	-	EFC-5	604	Unknown
C12-IRE1.1	Blood	2013	-	EFC-5	604	Unknown
C15-IR	Synovial fluid	2013	-	EFC-5	604	Unknown
C5-IR	Liver abscess	2010	GEH	EFC-6	ND	Unknown
C6-IR	Blood	2013	GEH, CIP, LVX	EFC-7	179	Unknown
C7-IRE1	Peritoneal fluid	2010	GEH	EFC-7	179	Guard
C19-IR	Blood	2013	GEH	EFC-7	179	Unknown
C28-IR	Blood	2013	GEH	EFC-7	179	Unknown
C9-IR	Endometrial biopsy	2013	-	EFC-8	ND	Unknown
C1-IR	Peritoneal fluid	2010	-	EFC-9	ND	Unknown
C2-IR	Blood	2013	-	EFC-10	ND	Unknown
C4-IR	Abdominal fluid	2010	GEH	EFC-11	ND	Unknown
C37-IR	Blood	2014	-	EFC-12	236	Traumatology
C49-IR	Blood	2014	-	E12	236	ICU
C41-IR	Synovial fluid	2014	-	EFC-12.1	236	Traumatology
C42-IR	Tissue abscess	2014	-	EFC-12.1	236	ER
C8-IR	Subphrenic abscess	2010	-	EFC-13	ND	Unknown
C10-IR	Blood	2013	-	EFC-14	ND	Unknown
C14-IRE1	Peritoneal fluid	2010	STH, CIP, LVX	EFC-15	ND	IM
C29-IR	Liver abscess	2010	STH, CIP, LVX	EFC-16.1	281	Unknown
C39-IR	Pericardial fluid	2014	-	EFC-16	281	ICU
C44-IRE1	Liver abscess	2014	-	EFC-16	281	ICU
C52-IRE1	Synovial fluid	2014	GEH, STH, CIP, LVX	EFC-16.1	281	Traumatology
C23-IR	Peritoneal fluid	2010	-	EFC-17	ND	Surgery
C24-IR	Tubo-ovarian abscess	2013	STH	EFC-18	ND	Unknown
C27-IR	Synovial fluid	2013	GEH, STH, CIP, LVX, CHL	EFC-19	ND	Traumatology
C25-IR	Blood	2013	-	EFC-20	ND	Surgery
C26-IR	Blood	2013	-	EFC-21	ND	Unknown
C20-IRE1	Liver abscess	2010	-	EFC-22	ND	Surgery
C21-IR	Synovial fluid	2013	GEH, STH	EFC-23	ND	Traumatology
C17-IR	Synovial fluid	2013	-	EFC-24	ND	Unknown
C18-IR	Abdominal fluid	2010	-	EFC-25	ND	ER
C36-IR	Skin abscess	2013	-	EFC-26	ND	Unknown

Abbreviations: ST, Sequence type; CC, Clonal Complex; ND, Not determined; GEH: gentamicin; STH: streptomycin; PEN: penicillin; CIP: ciprofloxacin; LVX: levofloxacin; CHL: chloramphenicol; IM, Internal medicine; ER, Emergency room; ICU, Intensive care unit.

**Table 2 pathogens-09-00142-t002:** Relevant characteristics of the 19 *E. faecium* strains isolated from human invasive infections at the HRS, Buenos Aires (Argentina).

Strain	Clinical Sample	Year	Antibiotic Susceptibility	*van* Genotype	PFGE-Type	ST	BAPS	HRS Ward
C47-IRE2	Abdominal fluid	2014	VAN, Q/D, PEN	*vanB*	EFM-1	25	2.3	Surgery
C48-IR	Blood	2014	VAN, TEC, Q/D, PEN	*vanA*	EFM-1	25	2.3	Surgery
C38-IR	Blood	2014	-	-	EFM-1	25	2.3	Traumatology
C30-IR	Blood	2013	VAN	*vanB*	EFM-1	25	2.3	ICU
C31-IR	Abdominal fluid	2010	LVX, PEN, AMP	-	EFM-2	ND	ND	ER
C32-IR	Intravesicular fluid	2010	-	-	EFM-2.1	ND	ND	Unknow
C34-IR	Sinovial fluid	2013	PEN	-	EFM-2	ND	ND	Unknow
C45-IR	Blood	2014	PEN, SAM	-	EFM-3	18	3.3	Traumatology
C52-IRE2	Sinovial fluid	2014	PEN, AMP, SAM	-	EFM-3	18	3.3	Traumatology
C16-IR	Blood	2013	PEN	-	EFM-4	ND	ND	ICU
C20-IRE2	Liver abscess	2013	-	-	EFM-4	52	7	Surgery
C35-IR	Peritoneal fluid	2010	-	-	EFM-4	52	7	ER
C3-IR	Blood	2013	-	-	EFM-5	ND	ND	Unknow
C7-IRE2	Peritoneal fluid	2010	GEH, Q/D	-	EFM-5	ND	ND	ER
C14-IRE2	Peritoneal fluid	2010	-	-	EFM-6	ND	ND	IM
C53-IR	Abdominal abscess	2014	VAN, TEC, GEH, CIP, LVX, PEN, AMP, SAM	*vanA*	EFM-7	792	3.1	ICU
C40-IR	Blood	2014	STH, PEN, AMP, SAM	-	EFM-8	19	7	Traumatology
C44-IRE2	Liver abscess	2014	VAN, TEC, LVX, CIP, PEN, AMP, SAM	*vanA*	EFM-9	792	3.1	ICU
C51-IRE2	Peritoneal fluid	2014	-	-	EFM-10	ND	ND	Surgery

Abbreviations: ST, Sequence type; VAN: vancomycin; TEC: teicoplanin; GEH: gentamicin; STH: streptomycin; PEN: penicillin; AMP: ampicillin; SAM: ampicillin/sulbactam; CIP: ciprofloxacin; LVX: levofloxacin; Q/D: quinupristin-dalfopristin; ND, Not determined; ICU, Intensive care unit; ER, Emergency room; IM, Internal medicine.

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
