# Peer review of "Detection of β-Lactamase-Producing Enterococcus faecalis and Vancomycin-Resistant Enterococcus faecium Isolates in Human Invasive Infections in the Public Hospital of Tandil, Argentina"

_pathogens, 2020, doi:10.3390/pathogens9020142_

Round 1

Reviewer 1 Report

minor English Editing is required through the text. My only concern is that why the authors conducted sampling between 2012 to 2014, so almost few years ago up to 6 to 8 years, and trying to publish the AMR data 6 years later. This may not be appropriate when interpreting the current AMR situation.

The validity of AMR research projects lies in their timely reporting.

Did the authors compare results of recently collected samples to the samples collected few years ago. 

Authors should include in the discussion a paragraph on the current situation in their locality with recent citations 

For me, and other peers in the field,  these results doesn't reflect the current situation.

Author Response

Reviewer 1:

Minor English Editing is required through the text. The manuscript has been revised by an English speaker.

My only concern is that why the authors conducted sampling between 2012 to 2014, so almost few years ago up to 6 to 8 years, and trying to publish the AMR data 6 years later. This may not be appropriate when interpreting the current AMR situation.  The validity of AMR research projects lies in their timely reporting.

Authors response: We disagree. AMR research comprises different areas including surveillance and also analysis of transmission and emergence of novel or unusual mechanisms. The main message of the paper was to highlight the occurrence of some important and unusual antimicrobial resistance profiles more than provide surveillance data. The manuscript has been modified to highlight the main message. Information about the epidemiology after 2014 is mentioned. 

Did the authors compare results of recently collected samples to the samples collected few years ago?

Authors response: See previous response.

Authors should include in the discussion a paragraph on the current situation in their locality with recent citations.

Authors response: We have included a comment about the absence of isolates carrying genes coding for resistance to glycopeptides or beta-lactamase.

For me, and other peers in the field, these results doesn't reflect the current situation.

Authors response: We do not understand the question of the reviewer. Epidemiology of resistant pathogens, including enterococci, vary in different geographical areas. This is explained in the introduction and discussion sections of the paper and also, in the cover letter of this manuscript where we highlight the novelty and relevance of the work (see also Freitas et al, JAC 2014).

Reviewer 2 Report

In general the manuscript is a good. The topics is really hot. 

I recommend a few modification.

Double check the English.

The discussion section, I would recommend cutting it to make an impact on the reader.

Conclusion need to be short and clear with 3 recommendation for other hospitals. 

Please recheck the References order.

Author Response

Reviewer 2:

Comments and Suggestions for Authors

In general, the manuscript is a good. The topics is really hot.

I recommend a few modifications.

Double check the English.

Authors response: The manuscript has been revised by an English speaker.

The discussion section, I would recommend cutting it to make an impact on the reader.

Authors response: Done

Conclusion need to be short and clear with 3 recommendation for other hospitals.

Authors response: Done

Please recheck the References order.

Authors response: Done

Round 2

Reviewer 1 Report

The manuscript requires to be revised for English by native speaker or scientific editor.

Author Response

The manuscript has been reviewed by an English native speaker.

The changes performed are marked in the manuscript with the Microsoft work "track changes" tool.

Aside from the English corrections some other changes were performed in the manuscript.